# Applications of Probiotic-Based Multi-Components to Human, Animal and Ecosystem Health: Concepts, Methodologies, and Action Mechanisms

**DOI:** 10.3390/microorganisms10091700

**Published:** 2022-08-24

**Authors:** Sonagnon Kouhounde, Kifouli Adéoti, Majid Mounir, Alessandro Giusti, Paulo Refinetti, Akaninyene Otu, Emmanuel Effa, Bassey Ebenso, Victor O. Adetimirin, Josep Mercader Barceló, Ousmane Thiare, Holy N. Rabetafika, Hary L. Razafindralambo

**Affiliations:** 1Laboratoire des Sciences Biologiques Appliquées, Université Aube Nouvelle, Bobo-Dioulasso 01 BP 234, Burkina Faso; 2Laboratoire de Microbiologie et de Technologie Alimentaire (LAMITA), Faculté des Sciences et Techniques, Université d’Abomey-Calavi, Cotonou 01BP 526, Benin; 3Department of Food Science and Nutrition, Biotransformations Laboratory, Hassan II Institute of Agronomy and Veterinary Medicine, Rabat Instituts, Rabat 10112, Morocco; 4CyRIC-Cyprus Research and Innovation Center Ltd., Nicosia 2414, Cyprus; 5REM Analytics SA, 1870 Monthey, Switzerland; 6Leeds Teaching Hospitals NHS Trust, Leeds LS1 3EX, UK; 7Foundation for Healthcare Innovation and Development (FHIND), Calabar 540271, Nigeria; 8Department of Internal Medicine, University of Calabar, Calabar 540271, Nigeria; 9Leeds Institute of Health Sciences, University of Leeds, Leeds LS2 9NL, UK; 10Department of Crop and Horticultural Sciences, University of Ibadan, Ibadan 200284, Nigeria; 11Molecular Biology and One Health Research Group (MolONE), University of the Balearic Islands, 07120 Palma, Spain; 12Department of Computer Science, Faculty of Applied Sciences and Technology, Université Gaston Berger de Saint Louis, Saint-Louis BP 234, Senegal; 13ProBioLab, Campus Universitaire de la Faculté de Gembloux Agro-Bio Tech/Université de Liège, B-5030 Gembloux, Belgium; 14BioEcoAgro Joint Research Unit, TERRA Teaching and Research Centre, Microbial Processes and Interactions, Gembloux Agro-Bio Tech/Université de Liège, B-5030 Gembloux, Belgium

**Keywords:** probiotics, prebiotics, synbiotics, postbiotics, human health, animal health, ecosystem health

## Abstract

Probiotics and related preparations, including synbiotics and postbiotics, are living and non-living microbial-based multi-components, which are now among the most popular bioactive agents. Such interests mainly arise from the wide range and numerous beneficial effects of their use for various hosts. The current minireview article attempts to provide an overview and discuss in a holistic way the concepts, methodologies, action mechanisms, and applications of probiotic-based multi-components in human, animal, plant, soil, and environment health. Probiotic-based multi-component preparations refer to a mixture of bioactive agents, containing probiotics or postbiotics as main functional ingredients, and prebiotics, protectants, stabilizers, encapsulating agents, and other compounds as additional constituents. Analyzing, characterizing, and monitoring over time the traceability, performance, and stability of such multi-component ingredients require relevant and sensitive analytical tools and methodologies. Two innovative profiling and monitoring methods, the thermophysical fingerprinting thermogravimetry–differential scanning calorimetry technique (TGA-DSC) of the whole multi-component powder preparations, and the Advanced Testing for Genetic Composition (ATGC) strain analysis up to the subspecies level, are presented, illustrated, and discussed in this review to respond to those requirements. Finally, the paper deals with some selected applications of probiotic-based multi-components to human, animal, plant, soil and environment health, while mentioning their possible action mechanisms.

## 1. Introduction

During the 20th century, probiotics have been recognized and used as beneficial live microorganisms only for human and animal health [1]. A few years later, the concept of probiotics has been extended and applied to plant growth and protection, soil fertilizing [2], and depolluting [3]. Now, probiotics are considered to be a biotool par excellence that offers multiple potential solutions in improving human life for food and nutrition security [4], disease prevention [5,6,7,8], and environment protection [9,10]. Beside high functional interests, probiotics also benefit the material status of natural [11], safe or “generally recognized as safe” (GRAS) and “qualified presumption of safety” (QPS) [12], and renewable, i.e., cultivable and inexhaustible, biomass sources [13].

Probiotics are live microorganisms, mainly belonging to lactic acid bacteria (*Lactobacilli*), *Bifidobacteria*, soil-based bacteria (*Bacillus* sp.), and yeast (*Saccharomyces* sp.) groups [14] with different species and strains from food and non-food sources [15]. Their multiple functionalities in promoting human and ecosystem health result from their capacity to control pathogens, reduce toxin and polluting substances, and increase nutrient bioavailability through three main action mechanisms [16]. These include, among others: (1) surface and nutrient competition ability against pathogens through cell wall macromolecular structures (e.g., S-proteins and exopolysaccharides) and secreted amphiphilic compounds (e.g., biosurfactants); (2) antimicrobial production (e.g., bacteriocins, antiviral agents, enzymes, antioxidant compounds); and (3) immunomodulation activity to immune cells. In fact, probiotics act as immunomodulators by increasing the growth of healthy components and restoring the normal gut ecological niche [17]. Probiotics can stimulate phagocytic activity, balance pro- and anti-inflammatory cytokines, enhance the production of immunoglobulin (IgA) by plasma cells, and generate bioactive peptides.

From a technological viewpoint, probiotics are commonly produced by culture in fermenters, and used as functional ingredients in formulated food and non-food products, including fermented foods and beverages, diet supplements, drugs, and biological and cosmetic products [18]. Rarely used in pure forms, probiotics are mostly prepared and formulated with a wide range of other functional compounds for preserving, and even enhancing, cell performance, viability, and stability [19,20]. These components include thermo- and cryo-protectors [21], prebiotics [22], and encapsulating agents [23,24], or another probiotic species for preparing multi-strain products. When probiotics are combined with prebiotics, represented mainly by oligosaccharides, phenolic compounds, or polyunsaturated fatty acids, which serve as selective nutrients for probiotics [25], the multi-component preparations are called synbiotics.

When microbial cells are inactivated by thermal processing (e.g., pasteurization, tyndallization, autoclaving), and no longer contain viable probiotic cells, or the preparation consists of probiotic fragments or their metabolites, the term postbiotics is used [26].

The concept of “probiotic-based multi-components” is therefore more appropriate to design the preparations or formulations of probiotic multi-strains, synbiotics, or postbiotics, which are all beneficial for human, animal, and ecosystem health [27,28] when used under adequate conditions. In terms of analysis and characterization, the use only of gold-standard methods (e.g., genotype and phenotype profiling) is insufficient for identifying and fingerprinting all components, particularly the “additives” included in these probiotic-based multi-components. Other analytical and physical chemical tools are needed for identifying and quantifying, for instance, prebiotics and protectants in such formulations [29].

The objective of this review paper is to (1) define the concept of probiotic-based multi-components; (2) describe two emerging analytical techniques for the global profiling of multi-component and multi-strain products, and monitoring the probiotic strain interactions over time; and (3) address relevant applications in human, animal, and ecosystem health, including plant, soil and environment.

## 2. Probiotic-Based Multi-Components

### 2.1. Probiotics and Synbiotics

Probiotic-based multi-components are products that contain either one strain or a mixture of strains [30] and additional functional compounds. These are mainly thermo- or cryoprotectants [20,21], mainly carbohydrates (e.g., lactose, maltose, trehalose, maltodextrins), proteins (e.g., skim milk powder, whey protein), minerals (e.g., Ca^++^), and other compounds (e.g., glycerol), but also antioxidants such as ascorbic acid, tocopherol, and flavonoids [31]. Such additional compounds are essentially used to protect microbial cells against the changes in stress parameters such as temperature, pressure, oxygen exposure, and relative humidity, which cause losses in cell viability during the drying process of the probiotic culture and the storage of the resulting dry material [32,33]. Other functional compounds found in probiotic preparations are encapsulating agents when coating/encapsulation techniques are used for ensuring sufficient survival rates of microbial cells, until they reach the human or animal gut [34]. Among the current components used in such processes are hydrocolloid biopolymers, mainly constituted by proteins and polysaccharides, either for bulk or single-cell encapsulation systems [24]. According to the physical state of the product (dry or wet form), other extra additives and functional components used in probiotic formulations, for instance, anticaking agents, minerals, buffers, etc., can influence the performance of probiotic cells in terms of metabolic activity and survivability [35].

When prebiotics are combined with mono- or multi-strain probiotic preparations, the multi-component systems are called synbiotics [36]. Prebiotics are selective nutrients for beneficial microorganisms harbored by the host, mainly carbohydrate compounds such as inulin and fructo- and galacto-oligosaccharides (FOSs and GOSs), which are considered safe food ingredients in the European Union [25]. Moreover, as renewable and sustainable resources with a relatively low cost, these biocomponents appear to be eco-friendly and economically advantageous for use in agro-food sectors. Some foods and plant-based materials such as soybean extracts, kojiglycosylceramides, grape extracts, tea polyphenols, and seaweed extracts can also stimulate the proliferation of beneficial microorganisms in the intestine, and are considered as prebiotics [37]. Although all current prebiotics are carbohydrates, some polyphenols compounds have emerged and mediate beneficial physiological effects by modulating the gut microbiota [38].

Synbiotics as mixtures of probiotics and prebiotics can be designed in complement to independently target the host microorganisms, or in synergism, for which the prebiotic is selectively utilized by the co-administrated microorganisms to achieve one or more health benefits [36]. Consequently, such probiotic-based multi-components can be designed using a multitude of combinations with a wide range of properties and functions. The use of synbiotics with regards to the synergistic aspect confers them economic and environmental assets. Numerous benefits of synbiotics to human health have been shown [36] in comparison with those of animal [39] and plant cases in nutrition and health [40]. Synbiotic-based multi-components also appear to be relevant for promoting both food nutritional security and sustainable agriculture, due to their roles as biofertilizers and biopesticides [2]. Table 1 lists some common synbiotics with their components in probiotics and prebiotics, in addition to their commercial names.

### 2.2. Postbiotics

The term “postbiotics” has been used for several years in different contexts, and the definition varies from one author to another, sometimes leading to confusion. Substances released by or produced through the metabolic activity of a microorganism that exerts a beneficial effect on the host, directly or indirectly, or substances of microorganism origin that confer beneficial effects to the host and differ from substances of a prebiotic nature, or non-viable probiotics, inanimate microorganisms and/or their components, paraprobiotics, and ghostbiotics, are among the terms currently used to name postbiotics [41,42]. A panel of scientist experts has declared that postbiotics are preparations of inanimate microorganisms and/or their components that confers a health benefit on the host [43]. The most recent concept of postbiotics uses the term “substances derived after the microorganisms are no longer alive, inanimate, dead or inactivated”, including intact cells or structural fragments of microbes such as the cell wall [26]. From a chemical viewpoint, postbiotics are heterogeneous multi-components of microbial metabolites from cell-free supernatants (e.g., supernatants of *L. acidophilus*, *L. casei*, *L. rhamnosus* GG, *S. cerevisae*, and *S. boulardii* cultures), or microbial fragments and lysates prepared by chemical and mechanical techniques such as sonication and heat treatments [42]. Metabolite-based postbiotics include exopolysaccharides (EPSs), e.g., β-glucan, antioxidant enzymes, short-chain fatty acids (SCFAs) such as acetic, propionic, and butyric acids, and vitamins; those from cell wall components are lipoteichoic acid, teichoic acids, peptidoglycan, cell surface proteins, and polysaccharides [44]. Some information on the potential applications of postbiotics for human health and their mechanisms of action related to antibacterial, antiviral, antioxidant, and anticancer activities have extensively been reviewed by different authors [45,46,47].

## 3. Innovative Profiling and Monitoring Methodologies

### 3.1. Thermophysical Profiling

#### 3.1.1. Principle

A thermal profiling and fingerprinting method combines both thermogravimetry and differential scanning calorimetry techniques (TGA–DSC). This is one of the most convenient techniques for characterizing and analyzing powder-based products [48,49,50]. This calorimetric-coupling technique has recently been used for the first time to fingerprint probiotic-based powder products [51]. Such an original approach provides unique qualitative and quantitative data related to the decomposition and transition phases for each probiotic powder sample, regardless of its complexity, by monitoring the changes in the material mass and energetic content under a constant temperature rise. Its potential validity has been shown by a comparative analysis using proteomics and intestinal permeability in vivo tests [52].

#### 3.1.2. Practice of TGA-DSC Coupling Method

The TGA-DSC thermal analysis consists in heating a sample while simultaneously monitoring its mass and energy content by gravimetric analysis and heat flow measurements, respectively. The TGA-DSC instrument includes a calorimeter, which is a special furnace for controlling and measuring temperature changes of the material, and a microbalance for mass measurements. A powder sample is weighted with high precision (±0.01 mg), and then deposited onto an aluminum crucible. Runs are performed by simultaneously monitoring the changes in the sample mass (m) and heat flow (HF) as a function of the linearly increasing temperature under defined conditions. The workflow, a synthetic example of thermal profiling, and a fingerprinting analysis of probiotic powder samples are provided in Figure 1. Such a method provides TGA (sample mass vs. time/temperature), DTG (TGA first derivative), and DSC (heat flux vs. time/temperature) curves from which a series of unique thermophysical data of each sample can be extracted (Table 2).

#### 3.1.3. Examples of Mono- and Multi-Strain Thermophysical Profiling

Illustrative examples of thermal profiling (Figure 2) and fingerprint data (Table 3) of representative mono-strains and various lots of multi-strain samples are unique. The reproducibility of the method has been demonstrated by an interlaboratory TGA profiles of a same multi-strain formulation [53].

In these examples, probiotic strains can be classified by their thermostability, that is, their main decomposition (*T*_max_) or transition (*T*_m_) temperatures, extracted from TGA or DSC curves, respectively. For instance, the probiotic sample of *B. subtilis* appears the most thermoresistant with the highest *T*_max_ = 351 °C and *T*_m_ = 353 °C, due to the bacterial cell capacity to sporulate [54], whereas *L. bulgaricus* is the least thermoresistant, depending on the structure and composition of the strain cell wall. Moreover, the residual material at 600 °C is an indicator of the purity and dose of the probiotic cells, which include in their components a variety of macromolecules, e.g., proteins and polysaccharides (higher *R*600), compared to the minor additional ingredients, often constituted by smaller compounds (lower *R*600) [55]. In addition, a variability in the same brand of multi-strain samples prepared in different countries (e.g., VSL#3) can be detected by TGA-DSC analysis [52] due to the difference in the carbohydrate-based protectant used (e.g., maltose, maltodextrin, starch). Figure 3a demonstrates the protectant impact (e.g., maltose) on the multi-strain formulation profile and fingerprint. Both major and minor ingredients in probiotic products are detectable through the present thermal profiling and fingerprinting method. Figure 3b illustrates the difference in thermal profiles between a formulated strain (*L. plantarum* ATCC 8014) containing lyoprotectants, compared to the same one in pure form.

#### 3.1.4. Advantages of the TGA-DSC Approach

The TGA-DSC profiling and fingerprinting method has many advantages compared to the gold standard approaches (e.g., phenotyping and genotyping) in quality control and authentication of probiotic powder-based products. It instantly reveals information on both probiotic strain and additional functional ingredients (e.g., cryoprotectants and antioxidants) in the formulation. In fact, these additional ingredients can also contribute to the functionality and performance of the product, and deserve careful attention. Moreover, the method is rapid, highly reproducible, sensitive, and adaptable to a high throughput analysis while requiring only a small amount of sample without pre-treatment. Besides the pure analytical aspects, this method can also provide some relevant fundamental information on the thermostability of the probiotic strains and the product formulation itself.

### 3.2. Advanced Testing for Genetic Composition (ATGC)

Advanced Testing for Genetic Composition (ATGC) can be considered as a “next generation genotyping” technique. It was built upon the use of Cycling Temperature Capillary Electrophoresis (CTCE) developed first in the Thilly Laboratory at MIT [56] and then at the Norwegian Radium Hospital.

ATGC is a targeted measurement technique that provides greater precision, specificity, and versatility than rtPCR, while being just as cost-effective and fast. It has been developed to translate discovery data from untargeted analysis (16S, shotgun metagenomics, etc.) into tests for routine use. Due to its speed and cost, ATGC is ideally suited for probiotic quality control (QC), microbiome monitoring in humans for personalized treatments, and soil microbiome analysis for precision agriculture.

#### 3.2.1. Origins of ATGC

Initially known as “Constant Denaturing Capillary electrophoresis” (CDCE), ATGC was first developed as a technique to detect and quantify genetic variations. In the 1990s it was successfully used to measure the mutational spectrum of γ-DNA polymerase [57], Pfu [58], and β-DNA polymerase [59]. Later, Cycling Temperature during the electrophoresis has been introduced as a means to improve the flexibility of the technique [60,61]. With this addition, the fragment selection process could be automated, and bio-informatic techniques were developed to enable the automatic identification of suitable fragments for separation using CTCE [62]. A first automated algorithm was then developed to design fragments that enable the complete analysis of human mitochondrial DNA [63]. This assay, once established, was successfully used in several lineage tracing applications [64,65,66,67,68]. Transitioning from CTCE to ATGC was achieved through the further development of algorithms and software for the design of assays in more complex genetic mixes (such as microbiomes).

#### 3.2.2. Principles of ATGC

The ATGC workflow has four steps, namely: (1) DNA extraction: the protocol depends on the samples being analyzed; (2) PCR: a classical PCR (not rtPCR) to amplify the target fragments; (3) CTCE: measuring the relative abundance of the different fragment variants in a sample; and (4) data analysis: combining the results of several fragments to produce an accurate profile of a sample’s composition.

CTCE enables an accurate and fast analysis of single nucleotide polymorphisms (SNPs) [61]. Because it relies on the separation of fragments after PCR, where fragments are almost identical (i.e., only one SNP difference), CTCE is not affected by PCR amplification biases, which is a significant drawback of rtPCR [69,70].

A single SNP changes the melting profile of a double-stranded DNA fragment. As the temperature is cycled throughout the electrophoresis, the two variants spend a different amount of time in open and closed configuration, which results in the separations shown on Figure 4.

The area under the peak is a direct estimate of the relative abundance of one variant relative to the other. Leveraging the ability to accurately quantify the relative abundance between two fragments differing by a single SNP, it is possible to design assays in which one primer specifically amplifies two species of micro-organisms. Combining several such primers, one can effectively “cover” a genus, for quantitative and precise profiling. Selecting such primers is not trivial and requires extensive bio-informatics. Using a property platform made available by REM Analytics (Monthey, Switzerland), it is possible to “map” a large number of genetic sequences. The results can be displayed in an interactive 3D map. From Figure 5, it is clear from this map that there are some comparisons that are more appropriate than others. It is easy to compare *L. gasseri* to *L. johnsonii*. However, a direct comparison between *L. gasseri* and *L. helveticus* would be impractical. It also appears that a grouping of *L. gasseri* with *L. johnsonii* to be compared with a group including *L. crispatus* and *L. helveticus* would be possible. However, the genetic map of Lactobacillus species and subspecies is different compared to that of Bifidobacterium (Figure 6). *B. breve* and *B. longum* form a clique, whereas *B. animalis* behaves almost like a separate genus, rather than a species within a genus. From this map, it is inferred that Bifidobacterium must be analyzed at the subspecies level, and that the species level analysis provides insufficient information.

From Figure 5 and Figure 6, it is possible to intelligently select primers of interest. Looking at the map in Figure 6, we see that individual *Bifidobacterium* species behave almost like genera, which have several sequences clustered together. Thus, further detailed maps are required for each individual species. Figure 7 shows a map of subspecies of *B. longum*. It can be seen that *B. longum* subspecies *longum* and *infantis* form distinct clusters, making the design of a subspecies primer effective and meaningful.

#### 3.2.3. Probiotic Quality Control Application

Guarantying the composition of multi-strain probiotics is a challenge in quality control [71]. Classical microbiology techniques based on selective culturing cannot distinguish between species and/or subspecies of the same strain. Furthermore, these techniques take several days to produce a result, days in which a batch cannot be released from production. Recently the use of antibody binding followed by flow cytometry [72] was proposed. However, polyclonal antibodies remain difficult to use in routine QC applications, and the transition from polyclonal to monoclonal antibodies is very expensive.

ATGC, with a fast assay design process, specificity, sensitivity, and precision, provides a valid alternative. Assays can be designed for species-level discrimination, and subspecies or strains. Several mixtures of species are first produced and measured by ATGC. These mixtures are then added together in equal volume, and the relative abundance measured in the original mix is used to predict the relative abundance in this second mix. The results of four samples run in triplicate show an average discrepancy between observed and predicted values of 2% with a 95% confidence interval of ±4% (Figure 8).

#### 3.2.4. Human Microbiome Application

ATGC is an ideal tool to bring human microbiome measurements from a “discovery phase”, in which the whole spectrum of possible micro-organisms is investigated with NGS, to a validation and diagnostic phase, in which specific assays are used to analyze subsets of the microbiome.

For example, in the vaginal microbiome, large-scale studies using sequencing techniques [73] have demonstrated that only a few species of *Lactobacillus* are present in the healthy vagina. Relying on these data, REM Analytics has developed a vaginal microbiome assay that produces quantitative and reliable results on the main micro-organisms species of the woman’s reproductive tract: *L. iners*, *L. crisptus*, *L. helveticus*, *L. jensenii*, *L. gasseri*, and *G. vaginalis*. Yoni Solutions (Monthey, Switzerland) is currently commercializing this assay, to be used in personalizing microbiome treatments for women suffering from recurrent vaginal infections or implantation failure.

Similarly, for baby gut microbiomes, existing evidence demonstrates that the ability of babies to metabolize different human milk oligosaccharides (HMOs) is determined by their population of Bifidobacterium [74]. In this case, subspecies of *B. longum* and *B. animalis* must be distinguished in order to better predict the HMO metabolism, which is in turn important for the recommendation of specific formula, or of probiotics to infants with digestive troubles. REM Analytics is currently commercializing a subspecies level Bifidobacterium assay for research uses only, and testing it as part of the NUTRISHIELD project (H2020 Grant agreement number 818110).

As the examples above illustrate, ATGC has great potential to be used in specific diagnostics when the target list of micro-organisms has been identified. It can also be used with data from metabolic studies, enabling the identification of key micro-organisms involved in a function and deemed important. Furthermore, it can be used in combination with probiotics to monitor specific changes induced by the probiotics.

#### 3.2.5. Soil Microbiome Application

Soil is a complex ecosystem in which several genera of bacteria interact with fungi and other organisms (not always micro). Obtaining a general profile of the soil microbiome is difficult, but can be achieved using NGS with a very large number of reads. However, such general profiles are not useful for precision agriculture. Chemical soil analyses measure specific micro-nutrients and elements of known relevance to crops. Similarly, microbiological analyses must develop specialized assays focusing on known functions for which a clear corrective strategy exists when deficiencies are identified.

Leveraging the significant advances in soil microbiome of recent years, it is possible to identify specific families of micro-organisms with known effects on soil. For example, the *Bacillus*, and *Bacillus*-related genera, are known beneficial microbes for soil, with several having well-characterized roles in crop growth. Furthermore, the number of such micro-organisms available on the market to be used as biofertilizers or biopesticides is growing fast. A list of species within the *Bacillus* family targeted by an existing ATGC assay is shown in Table 4. Each element has well-known functions, and is available for purchase on the market while being approved by regulatory authorities for use in agriculture. The same approach can also be developed to study mycorrhizal fungi for improving their use in agriculture.

#### 3.2.6. Limitations, Challenges, and Future Developments

ATGC, like most other microbiome profiling techniques, requires accurate DNA databases to design assays, in addition to high-quality reference material to calibrate and validate such assays. In the microbiome field, such material can be difficult to find. This is especially true in niche fields, such as women’s health, in which very few reference genomes are available for most bacteria. Furthermore, bacterial libraries such as ATCC or DSMZ have a limited number of strains isolated from women’s genital track that can be used as reference material.

ATGC is less susceptible to poor sequence databases than NGS or 16S, since assays can be calibrated once designed, and their specificity, precision, and detection profiles are well established once they are deployed. This means that having reliable reference material remains critical. In the agricultural field, there are several laboratories with extensive libraries of agricultural micro-organisms. This makes the acquisition of high-quality reference material straightforward.

In the area of probiotic quality control, reference material is available and has high quality. The precision, sensitivity, speed, and cost of ATGC makes it therefore a perfect tool for monitoring the relative abundance of strains in mixed probiotics, ensuring consistency across batches. Demonstrative results from internal research (not yet published) exist with the following types of probiotics:*Lactobacillus* and *Bifidobacterium*Spore-forming (e.g., *Bacillus*)Gram-negative (e.g., *Hafnia alvei*, *E. coli* Nissle)

## 4. Applications and Action Mechanisms

### 4.1. Human Health

Humans are a reservoir of diverse group of microbes, which together constitute the human microbiome. This microbiome plays a key role in modulating the host internal environment, defending the body against infectious organisms and maintaining the health of humans [84]. The emergence of superbugs resistant to commonly used antibiotics suggests that the development of simple, low-cost, and intrinsic approaches to maintaining health are crucial. Probiotics have been shown to supplement the host microflora and protect against various pathogens by improving gut barrier function and activate specific genes in host cells, thereby stimulating the host’s immune response [85]. The gut microbiota in humans exert systemic effects on host health, metabolism, nutrition, and the immune system, which accounts for their designation as a “hidden metabolic organ” [86]. The evolution of the gut microflora from birth through adulthood is influenced by diet, genetic make-up, lifestyle and age of the host, and use of antibiotics [84]. Imbalances in the composition and function of intestinal microbes, referred to as gut dysbiosis, are associated with various human diseases [87]. Consequently, manipulation of intestinal microbiota, through diets that stimulate beneficial bacteria colonization of the GIT [88] and the administration of probiotics [89], holds promise for maintaining health and treatment of diseases. A shift from the healthy symbiosis between the microbiota and the host to persistent dysbiosis has also been identified as a factor in obesity [90]. Probiotics supplement host microflora and provide protection against various enteric pathogens, with demonstrated remarkable functional attributes for meeting most of the basic human nutritional and clinical supplementation requirements [84]. Although probiotics are essentially beneficial gut microorganisms, some species of probiotics are not part of the normal human gut flora, and the beneficial effects observed are not the same for different strains [91]. The majority of probiotics are species from three genera, viz., *Lactobacillus*, *Bifidobacterium*, and *Saccharomyces* [92,93,94]. The most-used vehicles for prebiotic administration have been pharmaceutical formulas and dairy products [95].

Probiotics have antipathogenic, antidiabetic, anti-obesity, anti-inflammatory, anticancer, anti-allergic, anti-anxiety, and angiogenic properties in humans [84]. These properties have been successfully harnessed to induce remission in ulcerative colitis [96] and reduce both weight and blood pressure [97]. Probiotics have also been shown to ameliorate infection and antibiotic-associated diarrhea, *Clostridium difficile*–associated diarrhea, and conditions such as allergic rhinitis and atopic dermatitis (eczema) [97]. Further research is required into the long-term utility and safety of probiotics in various disease conditions. Probiotics have been used to treat gastrointestinal (GI) and non-GI conditions that include traveler’s diarrhea, acute infectious diarrhea in infants and children, antibiotic-associated diarrhea, irritable bowel syndrome, and ulcer and atopic dermatitis [98,99], with effects also exercised on the brain and central nervous system [84,100] and cancer cells [101]. The advantages of probiotics are, however, more clearly demonstrated for GI-related diseases [98]. Probiotics significantly reduce the risk for diarrhea [102,103], with greater effectiveness obtained in children than adults [104]. Their effectiveness at reducing the frequency of antibiotic-associated diarrhea has also been demonstrated [103,105,106]. Probiotic strains *L. fermentum* NCMB 52221 and 8829 have shown considerable potency for suppressing colorectal cancer cells in vitro [101]. Probiotics and their fermented metabolites (postbiotics) have shown activities that counter oxidative stress, a factor in ageing, in middle-aged mice [95].

Probiotics have been trialed as a therapy for necrotizing enterocolitis (NEC). NEC is a serious inflammatory gastrointestinal disease that primarily affects premature infants and has a mortality rate as high as 50%. A Cochrane review with a meta-analysis of twenty-four eligible trials involving preterm infants <37 weeks gestational age or <2500 g birth weight showed that enteral probiotic supplementation significantly reduced the incidence of severe NEC (≥stage 2) and no systemic infection with the probiotic organism was reported in the trials [107]. Putative mechanisms for probiotic action in the gut include: (1) upregulation of cytoprotective genes; (2) competition with other microbes; (3) downregulation of pro-inflammatory gene expression; (4) production of butyrate and other short chain fatty acids that nourish colonocytes; (5) support of barrier maturation and function; and (6) regulation of cellular immunity and Th1:Th2 balance [97]. Outstanding issues to address include determining which probiotic to use, whether infants <1000 g benefit, and how to mitigate the risk of probiotic sepsis.

Research in animal models has shown that important components in mammalian milk, such as sialylated galacto-oligosaccharides (GOSs), reduce the occurrence of NEC in neonatal rats [108]. This could account for the 6–10-fold lower NEC risk in breast-fed infants compared to formula-fed infants. Indeed, GOSs appear to shape the components of the intestinal microbiome. Complex polysaccharides such as β-glucan (BGL) with anti-inflammatory properties have also shown promise in boosting growth performance and intestinal epithelium functions in weaned pigs, and hens [109,110]. More research is needed into the applicability of BGL in managing gastrointestinal inflammatory conditions such as NEC in humans.

Probiotics also play an important role in dentistry, since oral infections are considered prime among other infections affecting humans. Effects of probiotics on oral health are both direct and indirect. Some probiotics produce digestive enzymes for metabolizing proteins and carbohydrates. Several randomized clinical trials have shown the possible benefits of probiotic dairy products for oral health in children, adolescents, adults, and the elderly [95]. These studies indicate a role for probiotics in caries prophylaxis. The incorporation of probiotics into dairy products is due to their ability to neutralize acidic conditions that promote dental caries, the irreversible microbial disease of the calcified tissues of the teeth [111], and suppression of the caries pathogen. Given consumers’ concerns about allergens and lactose intolerance in respect of traditional dairy food matrices, there have been concerted efforts towards the development of cereals, soy, fruits, vegetables, and chocolate as innovative food matrices [112,113]. Although most probiotics are safe, they may sometimes come with side effects that include constipation, flatulence, hiccups, nausea, infection, and rashes [98]. In recent years, probiotic strains have been considered a powerful ally in fighting and preventing respiratory tract infections [99]. Reduction in upper and lower respiratory tract infections from the administration of probiotics bacteria has been reported [114]. The increasing evidence between gut and lung function, resulting from gut–lung cross-talk, suggests a possible role for probiotics in the management of COVID-19, caused by the severe acute respiratory syndrome corona virus 2 (SARS-CoV-2) that assumed a pandemic status in February 2020 [115].

In terms of action mechanisms, probiotics are involved in the maintenance of health through diverse and interconnected mechanisms. Probiotics produce vitamins, enhance nutrient absorption, and possess enzymatic activities, such as β-glucurodinase, β-galactosidase, and bile salt hydrolase, among other, that are essential for the host metabolism [97,116]. Probiotics modify microbiota populations through the production of short chain fatty acids (SCFAs), which alter luminal pH, and antimicrobial compounds, such as bacteriocin [117]. Probiotics stimulate the production of mucin glycoproteins and secretory immunoglobulin A (sIgA) by globet and B cells, respectively [117]. Mucin is necessary for probiotic adhesion to the intestinal mucosa, while impairing the adhesion of pathogen bacteria. sIgA serves as the first line of defense in protecting the gut from pathogens. Probiotics further modulate the immune system by interacting with toll-like receptors, thereby leading to the activation of the innate immune response; activating T-regulatory cells; and increasing the production of anti-inflammatory cytokines and reducing proinflammatory cytokines [97,118]. The production of SCFAs also plays an important role in the immune and inflammatory responses [119]. Moreover, SCFAs activate insulin sensitivity and fatty acid oxidation in muscle, decrease lipolysis and increase adipogenesis in adipose tissue, and enhance satiety through the stimulation of intestinal glucagon-like peptide 1 secretion [97,120]. The relevance of the gut microbiome on distal organs has led to defining the terms gut–brain, gut–lung, and gut–skin axes, among others. In nervous system disorders, the production of neuroactive compounds plays a significant role [121]. Finally, the interplay between the gut microbiome and other host microbiomes (lung, skin) is thought to contribute to the development of respiratory and skin diseases, in addition to the mitigation of symptoms [122,123,124,125]. Figure 9 illustrates the action mechanisms of probiotics for promoting human health.

When probiotics are mixed with prebiotics, the resulting synbiotic preparation can develop either complementary or synergistic actions for human health [36]. Synbiotics help to manage several disease pathologies by targeting host gut microbiota, which play a crucial role in metabolism and protection against pathogens [126,127,128]. Synbiotics can act in balancing the gut microbiota by adjusting the Firmicutes-to-Bacteroidetes ratio, inhibiting harmful bacteria through direct antagonism (such as *Klebsiella*, *Escherichia coli*, and *C*. *difficile*) or excluding the latter by competitive adhesion, and accelerating the recovery to a healthy gut microbiome, e.g., by maintaining intestinal pH, producing important metabolites, and improving the gut mucosal barrier [129]. Health claims from clinical studies on synbiotics are linked to the gut health in order to treat inflammatory bowel syndrome (IBS) and inflammatory bowel disease (IBD), metabolic disease, and colorectal cancers. Other health claims relate to the treatment of systemic diseases such as allergies, hypocholesterolemia, osteoporosis, hepatic encelphalopathy; gut–brain axis diseases such as autism, depression, and anxiety [130]; and gut–lung axis respiratory diseases such as SARS-CoV-2 pathogenesis [131,132]. Some selected examples of synbiotic health benefits claimed from clinical studies are listed in Table 5. For further clinical trial results from studies conducted around the world, some databases are available online (https://www.clinicaltrials.gov/, accessed on 8 June 2021; https://www.clinicaltrialsregister.eu/, accessed on 1 January 2021).

An alternative strategy to achieve the human health benefits of probiotics is the administration of bioactive compound-based preparations derived from probiotics, i.e., postbiotics [43]. They have several advantages over probiotics in terms of safety and production costs. Postbiotic health benefits rely on their antimicrobial, antioxidant, anticancer, and immunomodulatory potentials [44]. The postbiotic compounds with antimicrobial activities include bacteriocins and other peptides, SCFAs, organic acids, and hydrogen peroxide. The probiotic antioxidant enzymes catalase, superoxide dismutase, and glutathione peroxidase reduce the concentration of reactive oxygen species. Bacteriocins, and, in particular, enterocin, have cytostatic and apoptotic effects against cancer cells. The health-promoting effects of postbiotics include favoring mineral absorption, relieving constipation, preventing intestinal inflammation, controlling glycaemia, and reducing food allergies. Recent clinical trials have demonstrated the impact of postbiotics in a wide age range of individuals. In infants, the inclusion of postbiotics in an infant formula modifies the fecal microbiome and metabolome towards a profile closer to that observed in breast-fed infants [142]. In middle-aged individuals, the intake of urolithin A, a postbiotic metabolite of ellagitannins, improves muscle performance [143]. Figure 10 summarizes the potential applications of postbiotics in promoting human health.

### 4.2. Animal Health

The main use of probiotics, prebiotics, postbiotics, and synbiotics in animal feeding is associated with their verified efficacy in modulation of the intestinal microbiota. Administration of probiotic strains, both individual and combined, may have a significant effect on absorption and utilization of feed, resulting in a daily increase in body weight and an increase in total body weight of various animals, including turkeys, chicken, piglets, sheep, goats, cattle, and horses. Probiotic microorganisms mostly intended for animals include *Lactobacillus* (e.g., *L. brevis*, *L. casei*, *L. crispatus*, *L. farciminisa*, *L. fermentum*), *Bifidobacterium* (e.g., *B. animalis*, *B. longum*, *B. pseudolongum*, *B. thermophilum*), other lactic acid bacteria (e.g., *Enterococcus faecalis* and *faecium*, *Lactococcus lactis*, *Leuconostoc citreum*, *Pediococcus acidilactici*) and some species of *Bacillus*, *Saccharomyces*, *Kluyveromyces,* and *Aspergillus* [144].

#### 4.2.1. Poultry

In recent decades, antibiotics have been widely added to poultry diets to maintain animal health and to prevent enteric diseases, which would impair productivity, increase mortality, and contaminate poultry products for human consumption [145]. Increased bacterial resistance to antibiotics in humans has caused an increase in public and governmental interest in eliminating sub-therapeutic use of antibiotics in livestock. Prebiotics and probiotics are two alternative approaches that can alter the intestinal microbiota and immune system to inhibit colonization by pathogens and therefore have the potential to prevent enteric diseases in poultry production [146]. The application of probiotics in poultry is strictly associated with the concept of competitive exclusion (CE) [147], which protect chicks against *C. jejuni*, *Listeria monocytogenes*, pathogenic *E. coli*, *Yersinia enterocolitica*, and *C. perfrigens* [148]. Furthermore, Lactobacillus-based probiotic cultures significantly reduced *Salmonella enteritidis* recovery in challenged neonatal broiler chicks [147].

The supplementation of synbiotics increased average daily gain but reduced the feed/gain ratio in broilers from 1 to 42 days of age. Moreover, dietary synbiotic inclusion increased breast yield and decreased abdominal fat in broilers [146]. By contrast, synbiotic supplementation lowered the cooking loss during heat treatment in a water bath, malondialdehyde (MDA) content, and total Cr content in the thigh muscle in broilers [146]. Regarding meat quality, lipid peroxidation is one of the most common causes of meat quality degradation in chicken, and can reduce nutritious value, produce taste and texture issues, and change the look of the meat [146]. By reducing MDA accumulation in the thigh muscle, synbiotic supplementation reduced meat lipid peroxidation, which may be favorable to meat quality and shelf life [146]. Supplementation of synbiotic to broilers’ diet at the dosage of 1.5 g·kg^−1^, composed of probiotics (*B. subtilis*, *Bacillus licheniformis*, and *Clostridium butyricum*) and prebiotics (yeast cell wall and xylooligosaccharide), can promote growth performance (increased weight gain and feed utilization efficiency), improve carcass characteristics (elevated breast muscle yield and reduced abdominal fat yield) and meat quality (increased pH24 h value in the breast muscle and decreased cooking loss in the thigh muscle), and reduce the product of lipid peroxidation (MDA) and Cr accumulations in the thigh muscle in broilers [146]. A probiotic strain, *L. plantarum* UY, inhibited the proliferation of influenza A (IFV) virus in the animal lung in a dose-dependent manner [149] and also stimulated the Thelper cells type 1 (Th1) immune response, resulting in higher synthesis of secretary IgA, leading to the removal of IFV from the lung.

#### 4.2.2. Pigs and Piglets

Pigs have specific immune and intestinal functions and weaning is a difficult period that can lead to a reduction in growth performance. Consequently, during this period, pigs are highly susceptible to pathogenic microorganisms, such as enterotoxigenic *Escherichia coli* (ETEC), causing enteric diseases [114,150]. Typical plant-based feeds contain 2.3–3.8% of xylans, which can reduce nutrient digestibility and induce a propitious environment for the growing of harmful bacteria, changing the gut associate microbiota in newly weaned pigs [151]. The application of LAB probiotics has been linked by several authors with beneficial effects in models of gastrointestinal infection using small animals. However, reports of the efficacy of probiotic treatment in alleviating intestinal infection in large animals remain scarce [152]. Synbiotics enhanced growth performance by reducing diarrhea, immune response, and oxidative stress in the jejunum [153]. In fact, exogenous enzymes (e.g., xylanase) have been successfully used to hydrolyze the β-1,4 backbone of xylan, releasing xylan oligosaccharides (XOSs) and, consequently, reducing the non-starch polysaccharide (NSP) content and the viscosity of digesta, increasing the digestibility of nutrients [154,155]. The use of the synbiotic mixture associated with 0.05% of herbal mixture showed an average weight gain [151].

The use of synbiotics promoted a smaller villus:depth crypt ratio, except when associated with 0.1% of herbal mixture, which was similar to the diet without additives [151]. Supplementation with a combination of a probiotic originating from anaerobic microbiota (bacteria—10^9^ CFU/mL, yeast—10^5^ CFU/mL, molds—10^3^ CFU/mL) and a prebiotic (malto-oligosacharides, sodium acetate, ammonia citrate) results in improved digestion of nutrients and reduced emission of harmful gases, and prevents bacterial infections during the weaning period [156].

#### 4.2.3. Ruminants

In recent decades, there have been considerable improvements in ruminant production, and these advances must continue in order to meet growing demands. Currently available data regarding effects of synbiotics on animal health are insufficient and require further studies. However, they clearly indicate the effective synergistic action of probiotics and prebiotics in the reduction of populations of bacterial gastrointestinal pathogens [157]. A method to manipulate the microbiota of the rumen during its growth period is to directly provide activators and/or probiotics, to establish a balance in the microbiota, which is more efficient during growth than in adults [158]. Recent studies suggest that integrating pre- and probiotics into ruminant feeds may improve various aspects of ruminant performance, mitigate disease, promote overall animal health and well-being, and reduce the environmental impacts of ruminant production. LAB and yeasts (*S. cerevisiae*) are used as ruminal activators/probiotics for their ability to affect the dynamics of the microbiota in the rumen and the way in which nutrients are decomposed [159]. It has been confirmed that individual or combined supplementation with *Saccharomyces cerevisiae* and *L. acidophilus* improved the growth performance of growing goats. Synbiotics provide these benefits by favorably modulating the microbial environment within the gastrointestinal tract of ruminant animals [160]. Although the mechanisms of action exerted by probiotics on ruminants are not well elucidated, dietary probiotic dosage to ruminants enhances development and maturation, growth and performance, milk production and composition, nutrient digestibility, feed efficiency, pathogen reduction, and mitigation of gastrointestinal diseases [161]. The addition of mix feed additives can affect the kinetics of gas and methane production, and not the level of pH or dry matter and organic matter digestibility. In dairy cows, probiotics containing live yeast boosted food intake, improved feed efficiency, average daily gain, and total weight, and increased milk yield and quality [162,163]. Probiotics and prebiotics, alone or in combination, in the diet of lambs finished under subtropical climate conditions may assist in reducing the unfavorable effects of high ambient heat load on dietary energy utilization [164]. Lambs supplied with probiotics and/or prebiotics showed higher gain efficiency and a lower ratio of observed-to-expected diet net energy compared to controls, with little influence on carcass features, whole cuts, or visceral mass. Table 6 lists the effects of some species of microbial pre-/pro-/synbiotics administrated under defined mode and dose conditions on various ruminant hosts.

#### 4.2.4. Fish

Application of biofriendly feed additives such as probiotics, prebiotics, and synbiotics are becoming popular dietary supplements with the potential to not only improve growth performance, but, in some cases, to also enhance immune competence and the overall well-being of fish and crustaceans [176]. Probiotics not only improve the health status of cultured animals but also help to ensure the safety of consumers [177]. The most commonly used are bacterial probiotics strains (e.g., *Bacillus* sp., *Lactobacillus* sp., *Bifidobacterium* sp., *Pseudomonas* sp., *Streptococcus* sp., *Arthrobacters* sp., *Microbacterium* sp., *Phaeobacter* sp., *Streptomyces* sp., *Enterococcus* sp., *Lactococcus* sp., *Micrococcus* sp., etc.), yeast probiotics (e.g., *S. cerevisiae*, *Debrayomyces hansenii*), micro-algal probiotics (e.g., *Tetrasehnis suecica*, *Spirulina platensis*), and bacteriophages probiotics (e.g., *Bacteriophages* sp.) [176]. The main roles played by probiotics in fish are: (1) improvement in growth and feed utilization of aquaculture species; (2) assist with the provision of essential nutrients and micronutrients such as vitamins and essential fatty and amino acids to the host species; (3) improvement in hemato-biochemical parameters as they allow a significant increase in the abundance of erythrocytes, but also elevate the number of white blood cells (WBCs), the latter enhancing non-specific immunity associated with neutrophils and macrophages; (4) improvement in fish culture systems through enhanced disease resistance, in addition to general health benefits to fish [176,178,179,180]. Moreover, under stressful situations, fish experience oxidative stress, causing lipid peroxidation and excessive malondialdehyde production (MDA) [181], which threaten the functionality of body tissues and cells and pose a risk of DNA damage [182]. A recent study showed that dietary *Pediococcus acidilactici* (PA) and pistachio hull-derived polysaccharide (PHDP) + PA used as a synbiotic resulted in low MDA levels in Nile tilapia, thus improving the antioxidative capacity [183]. Furthermore, synbiotics can improve the quality of water with beneficial influences on fish production. Gram-positive bacteria (e.g., *Bacillus* sp.) efficiently convert organic substances into carbon dioxide, whereas Gram-negative bacteria convert relatively more organic matter into biomass or bacterial slime [184]. For example, *Lactobacillus sp*. used as a probiotic simultaneously eliminated nitrogen and pathogens from polluted shrimp farms and then decreased fish mortality [185].

### 4.3. Plant and Soil Health

In the past decade, probiotics have been much applied to a wide range of industries such as aquaculture, food industries, human medicine, and agriculture. Some studies have been focused on successful practices, mechanisms of probiotics activities, and methods for optimizing the successful use of strains [186,187]. According to research results published in agriculture fields, the microbiome community known as probiotics can offer benefits to plant growth promotion, nutrient use efficiency (Figure 11), and pest and phytopathogen control [188,189]. Although many authors have demonstrated the interactions of probiotics with plants, a very little knowledge is available in the literature on the action mechanisms of prebiotics in the ecosystem. Results from research on forest ecosystems showed that fungal and bacterium communities can respond to environmental changes in accordance with host trees [190]. Vassilev et al. [191] demonstrated that *Piriformospora indica*, a beneficial microorganism for plants, can be used to produce a phosphate-enriched fermentation liquid through a repeated-batch fermentation process for improving soil fertility and plant productivity. In other work, it was proved that *Bacillus amyloliquefaciens* BChi1 and *Paraburkholderia fungorum* BRRh-4 can also increase growth and fruit yield of strawberries, and enhance their functional properties, such as the content of total antioxidants, carotenoids, flavonoids, phenolics, and anthocyanins [192]. In addition, other work demonstrated that microbial and biochemical indicators of soil health can be used to assess the ecological risk of soil. These results confirmed that soil respiration can be used for estimations of the soil ecological conditions and microbiological activity [193].

Basically, the ecosystem has been defined as a system of two components, constituting living organisms and inanimate or physical factors, respectively called biotic and abiotic components [195]. Biotic components comprise animals, microbial organisms, and plants, which are fed by nutrients, among which prebiotics and postbiotics constitute important elements. Prebiotics are molecules capable of stimulating both the intestinal microflora and other bacterial populations, including those growing in agricultural soils, by improving plant and soil health. Diverse sources of plant prebiotics such as fructo-oligosaccharides (FOSs), inulin, and galacto-oligosaccharides (GOS) are commonly cited, but xylans, pectins, and fructans are also among substrates used as carbohydrate-based prebiotics [196]. Information on postbiotics is very limited and associated research is quite recent. However, their role has been tested recently on animal, human, and plant health. It has been reported that postbiotics contribute to promoting plant growth by enhancing proliferation of shoots and rooting, and also having biocontrol effects on plants [194,197]. Limited studies on the effect of postbiotics on plants are available. Indeed, these derivate molecules from the plant probiotic microorganisms’ metabolism play mediating roles between probiotics and plants, acting as plant growth activators or in the defense of plants against certain stresses. Postbiotics interact via biochemical mechanisms with plant cellular membrane receptors through transduction of systemic signals, which leads to changes in plant gene expression at the plant level [198]. A large number of molecules obtained from probiotics activities act on plants, and significantly contribute to enhancing plant health performance, such as in terms of growth, yield, and resistance to stresses (biotic and abiotic).

### 4.4. Environmental Health

Probiotics play an important role as remediation agents, helping the host in responding to environmental changes. Certain genera also act as bioremediation or decomposing agents of hazardous substances [10], such as the case of a bacterial consortium (*Xanthomonadaceae*, *Brachybacterium* sp., *Bhargavaea* sp., *Gordonia* sp., *Thalassospira* sp., *Pseudomonas* sp., *Dietzia* sp., *Mesorhizobium* sp., *Cytophaga* sp., *Martelella* sp.), providing an innovative bioremediation approach. In this work, chitosan used as an encapsulated agent can stimulate the bacterial community of mangrove sediments [199]. Bioremediation in this case is based on the use of probiotics to degrade, reduce, or remove pollutants in the environment. The working mechanism of bioremediation involves several technical aspects such as biotransformation, biodegradation, mineralization, phytohydraulics, bioaccumulation, and biovolatilization, where the degrading microbes remove, transform, modify, and/or convert a complex compound of pollutants into simpler and less-toxic compounds. This bioremediation system has been successfully applied in cleaning contaminated sites [200], agricultural land [201], ground water [202], surface water [203], and sea water [204].

Conventional remediation strategies for most types of environmental contamination are not only expensive but also ineffective, especially in low contaminant concentrations [9]. Probiotics-assisted remediation has come forward as a cheap and easy alternative. Probiotics can act through four main action mechanisms divided into two categories, the binding and enzymatic degrading activities of toxins and pollutants, as summarized in Figure 12. LABs, yeasts, and soil probiotic bacteria are able to bind both organic toxins (e.g., mycotoxins and pesticides) and nonorganic pollutants (e.g., heavy metals) [3] through chelation, adsorption, and precipitation mechanisms. The nature and structure of the cell wall, surface macromolecules such as S-layer proteins and polysaccharides, and the environment conditions (e.g., pH and temperature), are among the factors that control the binding capacity of probiotics, which in turn depends on the surface hydrophobicity and electrical charge [3]. The binding mechanisms of toxins may also result from physical degradation of petroleum hydrocarbons [205]. Another mechanism is the production of enzyme-degrading toxins, such as organophosphorus-based pesticides [206], or proteolytic activity [207].

By combining probiotics with prebiotics, the resulting synbiotics are expected to develop a higher detoxifying capacity since prebiotics support the viability and functionalities of probiotics, which can improve the binding capacity of the mixture. The rare research work conducted on the synbiotic beneficial effects on bioremediation involved the combination of Lactobacilli and Bifidobacteria with inulin (prebiotic) for removing various substances, especially Pb [208].

Concerning postbiotics, it has been reported that both viable and nonviable LABs were able to bind toxic secondary metabolites such as fumonisin [209]. As the removal of mycotoxins involves an adhesion-type mechanism to cell wall components, rather than a covalent binding or binding by the metabolism, dead cells retain their binding ability [210]. Another case of postbiotic activity demonstrated in vivo was the removal of ochratoxin A from a liquid medium of foods using a mixture of sterilized yeast and a fermentation residue of beer (40:60). The binding action for toxin removal implied physical interactions with the cell wall since the changes in pH affected the degree of the activity [211].

## 5. Conclusions

As natural microbial-based and multifunctional materials, the probiotic-based multi-components described in this review article refer to bioactive agent mixtures derived from alive (probiotics and synbiotics) and nonalive (postbiotics) probiotic cells. In such a preparation, the microbial-derived components constitute the main functional ingredients, whereas prebiotics, protectors, stabilizers, and encapsulating agents are among the additional ingredients. Analyzing, characterizing, and monitoring the traceability, performance, and stability of such multi-component ingredients over time require convenient and sensitive measuring methodologies such as the TGA-DSC calorimetric technique and ATGC genetic strain analysis up to subspecies. Such methodologies are able to provide qualitative and quantitative profiling of both microbial- and non-microbial components in the preparation, as overviewed and discussed in this article. The applications of probiotic-based multi-components are not limited to human and animal health, but also extend to the promotion of the health of plants, the soil, and the environment, that is, our ecosystem health. Now, they can be used as biosupplements in food and feed, biopesticides and biofertilizers for promoting plant and soil health, and bioremediation and depolluting agents for cleaning up and protecting the environment. A large amount of effort is still needed to obtain insight into their multiple and complex mechanisms of action through unimaginable interactions among microbial and non-microbial components. The use of an efficacious combination of living and non-living entities from natural resources is among the promising “one health” approaches for tackling the disruption of human, animal, plant, and environmental health arising from climate change, urbanization, ocean acidification, and other calamities at the possible origin of emerging infectious diseases and epidemics worldwide.

## Figures and Tables

**Figure 1 microorganisms-10-01700-f001:**
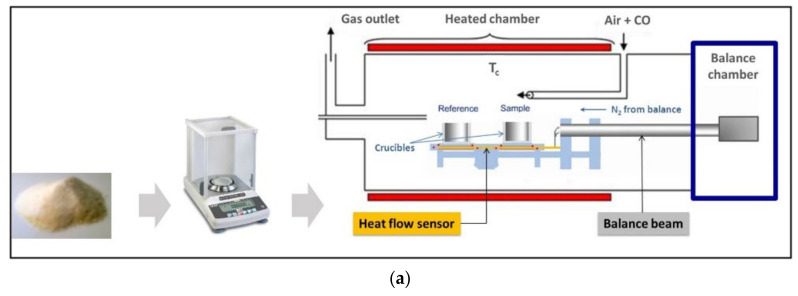
Main steps of the thermal probiotic powder profiling and fingerprinting experimentation: (**a**) from powder weighing to TDA-DSC analysis, and (**b**) output data.

**Figure 2 microorganisms-10-01700-f002:**
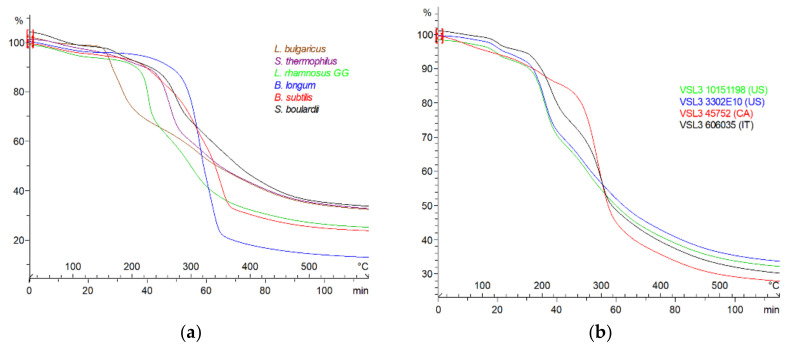
TGA curves of (**a**) mono-strains and (**b**) multi-strains.

**Figure 3 microorganisms-10-01700-f003:**
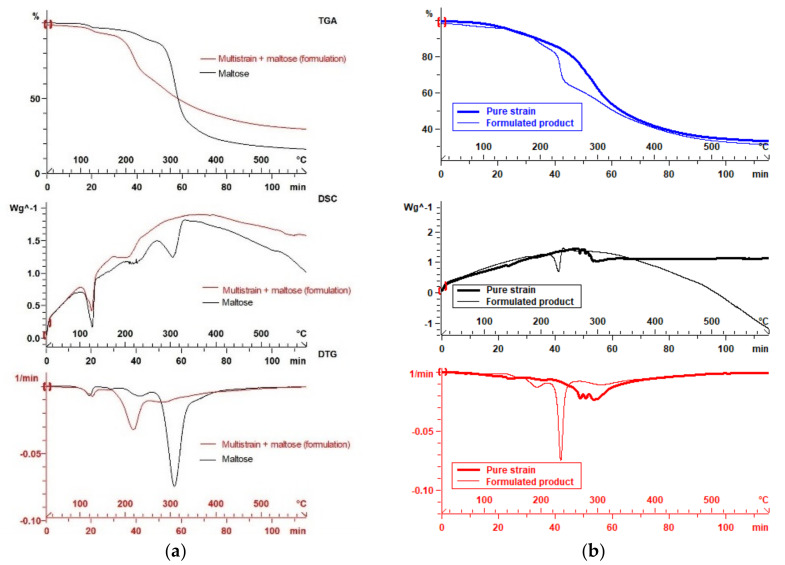
Thermal curves of (**a**) maltose-containing formulation vs. maltose and (**b**) pure vs. formulated *L. plantarum* ATCC8014.

**Figure 4 microorganisms-10-01700-f004:**
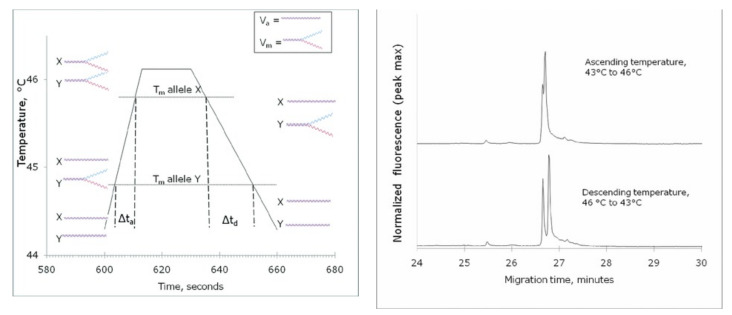
Basic principles of separation by CTCE “Reprinted with permission from Ref. [60]. 2022, John Wiley & Sons”.

**Figure 5 microorganisms-10-01700-f005:**
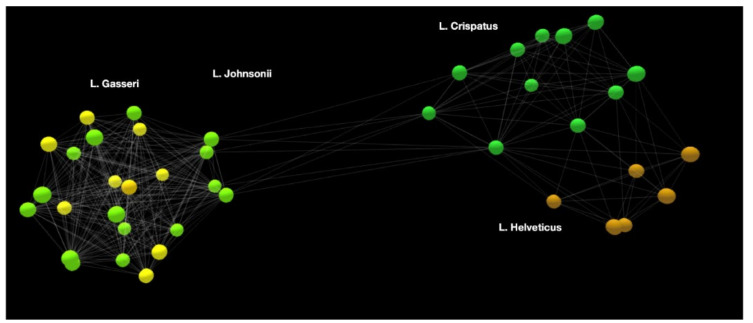
Example of genetic map of a subset of *Lactobacillus* species.

**Figure 6 microorganisms-10-01700-f006:**
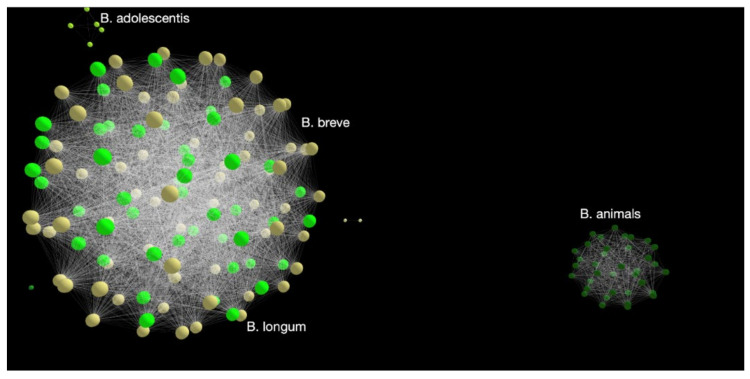
Example of a *Bifidobacterium* map.

**Figure 7 microorganisms-10-01700-f007:**
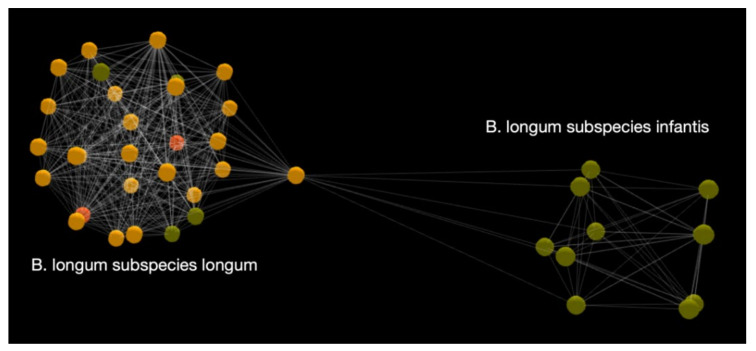
A genetic map of *B. longum* subspecies.

**Figure 8 microorganisms-10-01700-f008:**
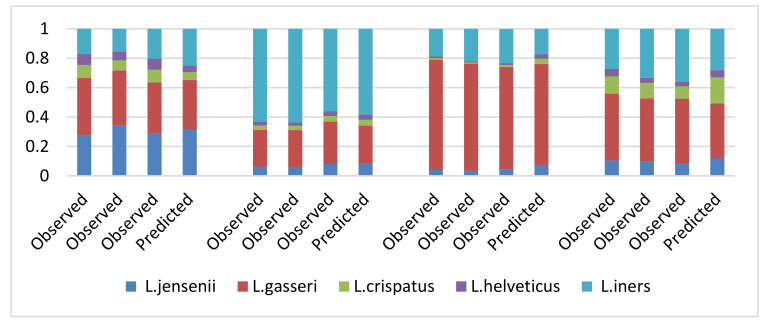
ATGC analysis of *Lactobacilli* mixtures.

**Figure 9 microorganisms-10-01700-f009:**
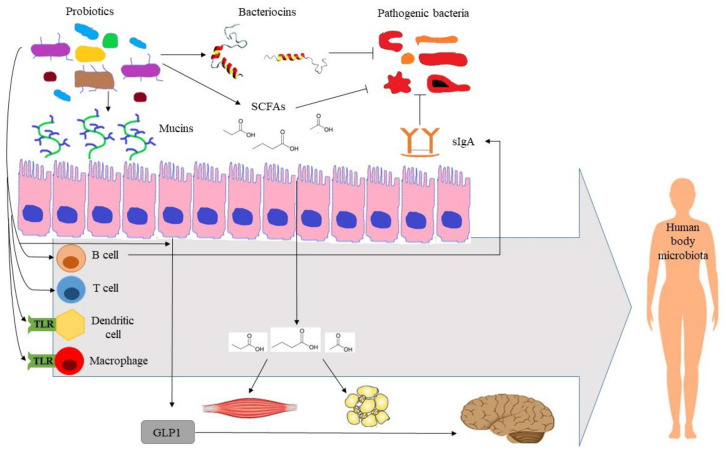
Illustration of action mechanisms of probiotics for promoting human health. GLP1: glucagon-like peptide 1, SCFAs: short-chain fatty acids, sIgA: soluble immunoglobulin A, TLR: toll-like receptor.

**Figure 10 microorganisms-10-01700-f010:**
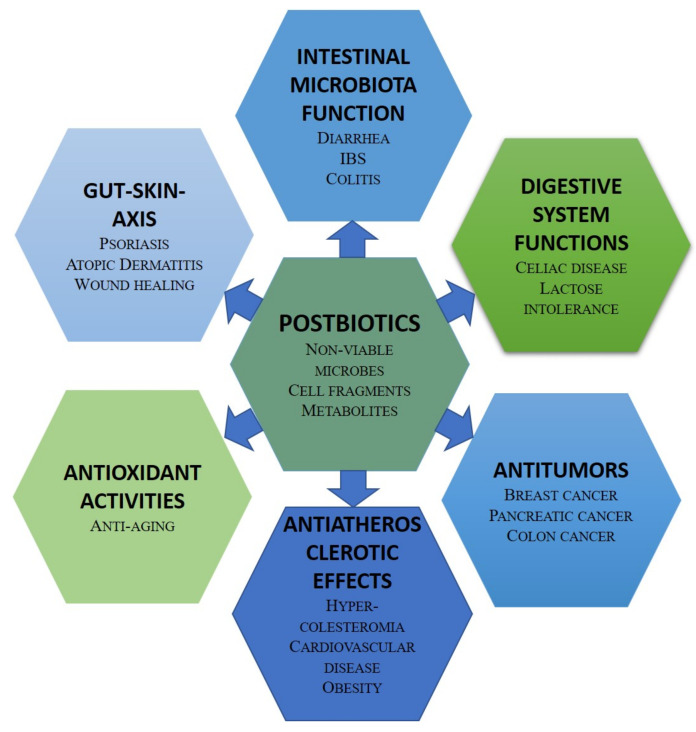
Potential applications of postbiotics in human health.

**Figure 11 microorganisms-10-01700-f011:**
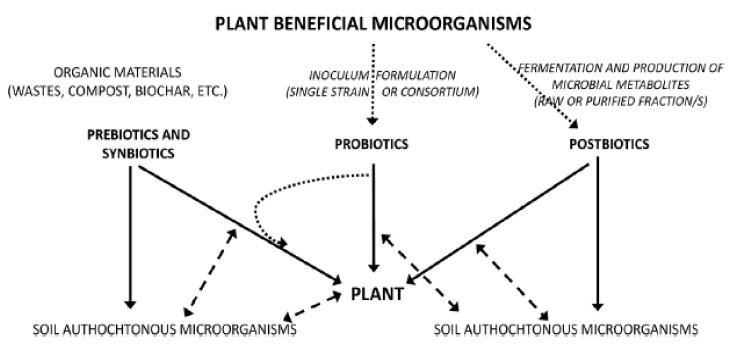
Diagram showing the three strategies for microbial soil–plant management based on prebiotics, probiotics, and postbiotics approaches. Legend: Full lines show the direct effect, dashed lines show the interactions, dotted lines show the formulation/production processes [194].

**Figure 12 microorganisms-10-01700-f012:**
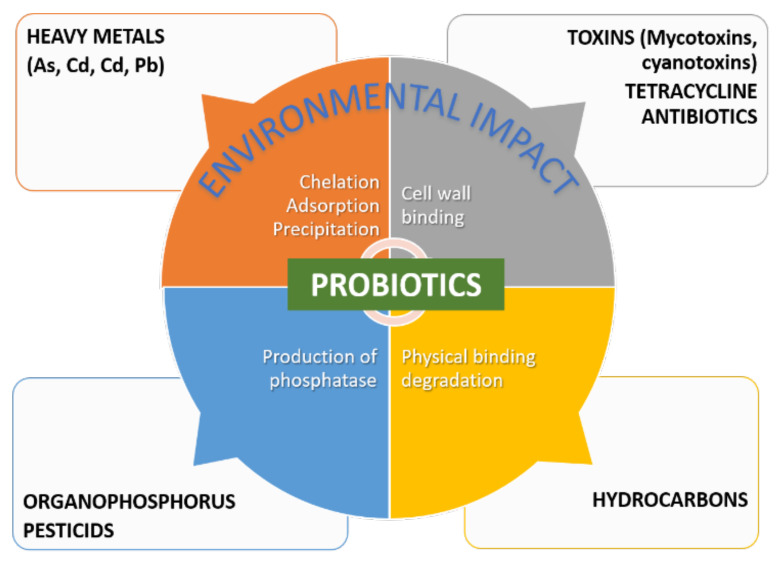
Main mechanisms of action of probiotics in remediation.

**Table 1 microorganisms-10-01700-t001:** Some examples of synbiotics, their probiotic and prebiotic components, and trade names.

Synbiotics		
Probiotics	Prebiotics	Products
*Lacticaseibacillus paracasei* YIT 9029 (strain Shirota: LcS) *Bifidobacterium breve* YIT 12272 (BbrY)	GOS	Super Synbiotics LBG-P (Yakult Honsha Co., Ltd., Tokyo, Japan)
*Streptococcus thermophilus* *Lacticaseibacillus rhamnosus* *Lactobacillus acidophilus* *B. infantis* *B. lactis*	FOS + Ascorbic acid	Probiotical (Laboratoires Phacobel Belgium, Soheit-Tinlot, Belgium)
*B. breve*	Short chain scGOS Long chain lcFOS	Danone Nutricia Research, Utrecht, The Netherlands
*Bacillus coagulans* MTCC 5856	Cranberry fiber	Lactocran (Sabinsa Corporation, Piscataway, NJ, USA)
*B. lactis, B. breve, B. infantis, B. longum*	XOS	PrebioMed™ XOS (designs for health, Suffield, CT, USA)
Mix of *Bifidobacteria* and *Lactobacilli*	Whole fruit Indian Pomegranate (Punica granatum) (>40% Polyphenolic + Phenolic Bioactives)	DS-01 (Seed Health, Los Angeles, CA, USA)
*L. acidophilus* *Lacticaseibacillus casei, L. rhamnosus* *Lactiplantibacillus plantarum*, *S. thermophilus* *B. longum*	Oat Bran (10% β-Glucan fiber) Organic Red Beetroot Inulin from Organic Chicory Root	Beta Glucan Synbiotic (BioImmersion Inc, Bellevue, WA, USA)
*B. animalis,* *Enterococcus faecium,* *Limosilactobacillus reuteri* *Ligilactobacillus salivarius* *Pediococcus acidilactici*	Inulin	PoultryStar^®^ (ME BIOMIN GmbH, Niederösterreich, Austria)
*Enterococcus faecium*	FOS	Biomin^®^ IMBO (ME BIOMIN GmbH, Niederösterreich, Austria)
*L. acidophilus* *L. casei* *L. salivarius* *L. plantarum, L. rhamnosus* *Levilactobacillus brevis* *B. bifidum* *B. lactis* *S. thermophilus*	Inulin	Synbiotic poultry (Vetafarm, Wagga Wagga, Australia)

**Table 2 microorganisms-10-01700-t002:** Main thermophysical quantities from the thermal profiling of probiotic-based products.

Thermophysical Quantities	Meaning	Source
***T*_max_** [°C]	Maximum temperature of main decomposition (first derivative or DTG main peak)	TGA
***V*_max_** [Kmin^−1^]	Maximum rate of main decomposition (first derivative or DTG main peak)	TGA
***T*_m_** [°C]	Temperature of phase transition of main decomposition	DSC
**Δ*H*_m_** [J/g]	Enthalpy of phase transition of main decomposition (area of the curve)	DSC
***R*600** [%]	Black carbon and mineral compounds at the end of the temperature scan (600 °C)	TGA

**Table 3 microorganisms-10-01700-t003:** Thermophysical data of mono-strain and multi-strain (VSL#3) samples of probiotic formulations.

Probiotic Formulations	*T* _max_	*V* _max_	*T* _m_	Δ*H*_m_	*R*600
	[°C]	100 × [min^−1^]	[°C]	[J/g]	[%]
**Mono-strains**					
*B. subtilis*	350.7 ± 0.7	3.93 ± 0.11	352.9 ± 0.9	−23.0 ± 3.0	24.0 ± 1.1
*B. longum*	314.3 ± 0.1	7.04 ± 0.07	310.9 ± 0.4	−5.1 ± 1.7	12.8 ± 0.3
*S. boulardii*	275.4 ± 0.3	2.60 ± 0.03	279.5 ± 0.4	33.3 ± 2.0	33.4 ± 0.3
*S. thermophilus*	264.6 ± 0.1	4.01 ± 0.03	273.2 ± 1.0	12.4 ± 1.2	32.7 ± 0.2
*L. rhamnosus GG*	228.2 ± 0.5	6.44 ± 0.06	228.1 ± 1.0	26.1 ± 4.0	24.5 ± 0.3
*L. bulgaricus*	165.2 ± 0.0	3.21 ± 0.04	165.6 ± 0.0	−97.9 ± 1.3	32.2 ± 0.1
**Multi-strains VSL#3**					
*Lot 606035*	301.2 ± 0.2	2.97 ± 0.02	305.8 ± 1.7	31.8 ± 4.5	30.1 ± 0.0
*Lot 45752*	294.1 ± 0.9	4.24 ± 0.03	290.0 ± 3.0	70.0 ± 8.3	28.4 ± 0.6
*Lot 10151198*	206.9 ± 2.1	3.00 ± 0.01	196.4 ± 4.5	−33.5 ± 2.8	32.9 ± 0.9
*Lot 3302E10*	204.3 ± 3.0	3.13 ± 0.11	193.7 ± 4.6	−41.2 ± 11.5	33.4 ± 0.2

**Table 4 microorganisms-10-01700-t004:** A list of Bacillus family species and their application in agriculture for ATGC assay.

Species	Known Function	Reference
*Bacillus subtilis*	Solubilize soil (phosphorus), enhance nitrogen fixation, and produce siderophores that promote its growth and suppresses the growth of pathogens	[75]
*Cytobacillus firmus*	Nematode antagonist	[76]
*Bacillus thuringiensis*	Known pesticide activities	[77]
*Bacillus mucilaginosus*	Solubilize potassium from minerals in soil so that plants, such as food crops, are able to use it	[78]
*Paenibacillus polymyxa*	Fix nitrogen, making it available to plants	[79]
*Bacillus cereus*	Regenerate contaminated soils and promote mycorrhizae growth	[80]
*Bacillus amyloliquefaciens*	Prevent a series of bacterial pathogens in crops	[81]
*Bacillus pumilus*	Promote plant growth, fix nitrogen, and prevent the germination of several fungal pathogens on crop roots	[82]
*Bacillus megaterium*	Solubilizes phosphates, and promotes plant growth through signaling	[83]

**Table 5 microorganisms-10-01700-t005:** Example of human health benefits of synbiotics claimed from clinical studies.

	Health Outcomes	Probiotic Strains	Prebiotics	Reference
Gut Intestinal tract	Treatment of overweight and metabolic syndrome	*L. casei* PXN 37, *L. rhamnosus* PXN 54, *S. thermophilus* PXN 66, *B. breve* PXN 25, *L. acidophilus* PXN 35, *B. longum* PXN 30, *L. bulgaricus* PXN 39	FOS	[133]
	Treatment of IBS	*Bacillus coagulans*	FOS	[134]
	Acute diarrhea	*L. acidophilus*, *L. rhamnosus*, *B. bifidum*, *B. longum*, *Enterococcus faecium*	FOS	[135]
	Colorectal cancer	*B. lactis*	Resistant starch	[136]
Kidney	Treatment of chronic kidney disease	*L. casei*, *L. acidophilus*, *L. bulgaricus*, *L. rhamnosus*, *B. breve*, *B. longum*, *S. thermophilus*	FOS	[137]
Liver	Treatment of non-alcoholic fatty liver disease Prevention of infections after liver transplant	*B. longum* *L. acidophilus*	Inulin HP	[138]
Lung	Reduction of viral respiratory infections in asthmatic children	*L. casei*, *L. rhamnosus*, *S. thermophilus*, *B. breve*, *L. acidophilus*, *B. infantis*, *L. bulgaricus*	FOS	[139]
Skin/derm	Treatment of atopic dermatitis	*L. salivarius* PM-A0006	FOS	[140]
Brain	Improvement in mental health in hemodialysis patients	*L. acidophilus* strain T16, *B. bifidum strain* BIA-6, *B*. *lactis strain* BIA-7, *B*. *longum strain* BIA-8	Equal mix of FOS, GOS and inulin	[141]

**Table 6 microorganisms-10-01700-t006:** Effect of some microbial pre/pro/synbiotics on ruminant production.

Ruminant Host	Pre-/Pro-/Synbiotics	Mode of Administration/Dose	Effect	Reference
**Dairy cows**	*L. casei* and *L. plantarum*	Combination of both in the feed (50 g/day)	Increases the milk production and the contents of milk immunoglobulin G, lactoferrin, lysozyme and lactoperoxidase	[165,166]
	*Propionibacterium* spp. and *S. cerevisiae*	Oral administration, mixed in feed	Improves the feed conversion rate, milk production and dry matter intake	[162]
	*S. cerevisiae*	Oral administration, mixed in feed (0.2 g/day)	Improves the feed conversion rate, milk production and dry matter intake	[167]
	*Fructo-oligosacchrides (FOSs)* and Mannan-oligosaccharides (MOSs)	Oral administration	Provides specific bacteria with a competitive advantage in the gut.	[168]
	*Enterococcus faecium* + lactulose	Oral administration 10^9^ colony forming units (probiotics) + 1–3% dry matter (prebiotics)	Decreases the ileal villus height, the depth of the crypts in the cecum, and the surface area of lymph follicles from Peyer’s patches	[169]
	*Streptococcus faecium* + Mannan-oligosaccharide (MOS)	Oral administration (0.6 kg/day)	Improves fecal consistency and reduced the fecal score of calves	[170]
	*S. cerevisiae* strain 1026 + Inulin	Oral administration (probiotic 5 g + prebiotic 6 g)	Impacts positively the development of morphological structures of digestive systems	[171]
**Goats**	*L. reuteri*, *L. alimentarius*, *Enterococcus faecium* and *B. bifidum*	Oral administration, resuspended in milk (1 mL/two feeds per day)	Improves the microbial environment and intestinal health, in addition to the acid profile of milk, with an increase in unsaturated fatty acids, mainly linoleic, linolenic and conjugated linoleic acids, and a decrease in the atherogenic index	[159]
	Inulin, fructo-oligosaccharide, galacto-oligosaccharide and xylo-oligosaccharide	0.4 to 0.6% in milk	Antioxidant activity and promotes the development of functional goat milk	[172]
**Sheep**	*S. cerevisiae* and two strains of rumen-derived *Diutina rugosa*	Oral administration (100 mL)	Stabilizes the ruminal pH, improves the richness of rumen microflora, relieves acidosis and inflammation, and prevents subacute ruminal acidosis	[173,174]
	*Propionibacterium* P63, *L. plantarum* and *L. rhamnosus*	Intraruminal cannula (2 g/day)	Stabilizes the pH of the rumen and prevents acidosis	[173,174]
	Mannan-oligosaccharide and *b-glucans*	Intraruminal cannula	Additive effects on digestion and fermentation	[175]
**Cattle**	*Enterococcus faecium* strain 26 and *Clostridium butyricum* strain Miyari	Oral administration	Reduces the ruminal pH and the concentration of lactic acid in the ruminal fluid, thus preventing acidosis	[159]
	Cellooligosaccharide (CE), Mannan-oligosaccharides (MOSs) and fructo-oligosaccharides (FOSs) such as Galactosyl-lactose in combination with spray-dried bovine serum	Oral administration of supplemented milk replacer	Reduces the incidence and severity of enteric disease and modulate the intestinal bacterial community in calves	[165]

## Data Availability

Not applicable.

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
