# Peer review of "Applications of Probiotic-Based Multi-Components to Human, Animal and Ecosystem Health: Concepts, Methodologies, and Action Mechanisms"

_microorganisms, 2022, doi:10.3390/microorganisms10091700_

Round 1
Reviewer 1 Report
This manuscript is extremely interesting and worthy of publication. The topic is innovative because it presents the approach to probiotic preparations more from chemical than microbiological terms. I believe that this manuscript must be allowed to be published in the MICROORGANISMS journal, but earlier the authors should improve it a bit and render it a bit. My suggestions are as follows:
· The text should be taken into account and use the current lactobacillus taxonomy (see: http://lactobacillus.ualberta.ca/). This is especially important when discussing the ATGC technique - not on the example of typical Lactobacillus (e.g., L. gasseri, L. johnsonii, and L. helveticus, see fig 5), but those lactobacilli that are currently classified to other bacterial types, e.g., Lacticaseibacillus casei, Lacticaseibacillus rhamnosus, Lactiplantibacillus plantarum subsp. plantarum, Levilactobacillus brevis, Limosilactobacillus fermentum or Limosilactobacillus reuteri. What is the use of this method for probiotic spore forming bacterial strains (e.g., Bacillus clausii) or gram-negative bacteria (e.g., E.coli Nissle)? And how does this method work in the case of animal probiotics, where the composition of the microflora is very diverse in terms of microbiologically?
· Table 4 – I think that such a list should be made based on clinical trial databases (see: https://www.who.int/observatories/global-observatory-on-health-research-and-development/resources/databases/databases-on-processes-for-r-d/clinical-trials, https://www.clinicaltrials.gov/, https://www.clinicaltrialsregister.eu/, https://www.isrctn.com/).
· Section „4.1. Human health” – What about NEC, GOS and newborns, especially premature babies? Are there any data on the use of beta-glucan?
· Table 5 – In my opinion, the data should also take into account the addition of prebiotics.
· Section „4.3. Plant & Soil health” and „4.4. Environmental health” – What prebiotics are examined or used in these applications?
Author Response
Dear Reviewer 1,
Thank you very much for your pertinent comments/remarks.
Please find below the revisions made to the first version of our manuscript according to your suggestions/comments.
We hope that this revised version will be suitable for the publication of our manuscript to this Special Issue for Microorganisms.
We look forward to hearing from you.
Sincerely,
RAZAFINDRALAMBO Hary (PhD)
Head of ProBioLab
Leader of WP4 - URBANE project (Horizon Europe 2021-2027)
https://urbane-project.eu/Projects/URBANE.html
--------------------------------------------------
TERRA Teaching & Research Centre
Gembloux Agro-Bio Tech
University of Liege
BELGIUM
Email: h.razafindralambo@uliege.be
Phone: +32 81 62 26 23
REVIEWER 1
• Lactobacilli taxonomy:
New nomenclatures are used for the first appearing of each probiotic genus & species (Table 1) Lines 144-145
• Use of ATGC for probiotic spore or gram (-) bacteria:
A phrase has been added to complete the applications of ATGC Lines 399-403
• Use of ATGC for animal probiotics:
Here is the answer to the question:
"The co-author from REM analytics currently has a major project on animal microbiome (goat mostly): Code-Refarm, https://coderefarm.eu . In this context, we demonstrated the ability of ATGC to analyse the microbiome of goat milk. It could also be used to analyse animal probiotics, however extensive results are not yet available. The microbiome variability is not a major issue for the method, since it focusses on target. Although selection of the target requires work for each species of animal being studied".
• Table 4 to be made based on clinical trial databases
Here are the answer and revision made to this question. The former Table 4 (Now Table 5) has not be changed since it has already included some clinical trial results from literature. However, we have added a phrase along with the links of websites for the clinical trial databases such as
- https://www.clinicaltrials.gov/
- https://www.clinicaltrialsregister.eu/
"For further clinical trial results from studies conducted around the world, some databases are available online (https://www.clinicaltrials.gov/; https://www.clinicaltrialsregister.eu/" (Line 536-538).
• Section 4.1. Human Health - What about NEC....
New inputs (Line451-472), including references, as below have been added in this section according to the Reviewer's comments:
"Probiotics have been trialled as a therapy for Necrotizing enterocolitis (NEC). NEC is a serious inflammatory gastrointestinal disease that primarily affects premature infants and has a mortality rate as high as 50%. A Cochrane review with meta-analysis of twenty-four eligible trials involving preterm infants < 37 weeks gestational age or < 2500 g birth weight showed that enteral probiotics supplementation significantly reduced the incidence of severe NEC (≥ stage 2) and no systemic infection with the probiotics organism was reported in the trials.1 Putative mechanisms for probiotic action in the gut include: 1) upregulation of cytoprotective genes; 2) competition with other microbes 3) downregulation of pro-inflammatory gene expression; 4) production of butyrate and other short chain fatty acids that nourish colonocytes; 4) support of barrier maturation and function; 5) regulation of cellular immunity and Th1:Th2 balance.2 Outstanding issues to address include determining which probiotic to use, whether infants <1,000 g benefit and how to mitigate the risk of probiotic sepsis.
Research in animal models has shown that important components in mammalian milk such as sialylated galacto-oligosaccharides (GOS) reduce the occurrence of NEC in neonatal rats.3 This could account for the 6–10-fold lower NEC risk in breast-fed infants compared to formula-fed infants. Indeed, GOS appear to shape the components of intestinal microbiome. Complex polysaccharides such as β-glucan (BGL) with anti-inflammatory properties have also shown promise in boosting growth performance and intestinal epithelium functions in weaned pigs, and hens.4,5 More research is needed into the applicability of BGL in managing gastrointestinal inflammatory conditions such as NEC in humans.
Additional references:
1. AlFaleh K, Anabrees J. Probiotics for prevention of necrotizing enterocolitis in preterm infants. Cochrane Database of Systematic Reviews 2014, Issue 4. Art. No.: CD005496. DOI: 10.1002/14651858.CD005496.pub4.
2. Plaza-Diaz J, Ruiz-Ojeda FJ, Gil-Campos M, Gil A. Mechanisms of action of probiotics. Advances in Nutrition, Volume 11, Issue 4, July 2020, Page 1054.
3. Autran, C., Schoterman, M., Jantscher-Krenn, E., Kamerling, J., & Bode, L. (2016). Sialylated galacto-oligosaccharides and 2′-fucosyllactose reduce necrotising enterocolitis in neonatal rats. British Journal of Nutrition, 116(2), 294-299. doi:10.1017/S0007114516002038.
4. Vetvicka V, Oliveira C. β(1-3)(1-6)-D-glucans modulate immune status in pigs: potential importance for efficiency of commercial farming. Ann Transl Med. 2014;2(2):16. doi:10.3978/j.issn.2305-5839.2014.01.04.
5. Jacob JP, Pescatore AJ. Barley β-glucan in poultry diets. Ann Transl Med. 2014;2(2):20. doi:10.3978/j.issn.2305-5839.2014.01.02
• Table 5 - In my opinion, ....
The former Table 5 (now Table 6) has been updated by including pre- and synbiotics applications to ruminants (Line 660-661).
• Section 4.3. "Plant & Soil health"
Section 4.3. Plant & Soil health has been updated by including prebiotics with probiotics applications (Line 718-725).
• Section 4.4 "Environmental health"
Here is the answer to the question: Inulin is the prebiotic used in these applications (Line 771-774)
Reviewer 2 Report
The manuscript Applications of Probiotic-Based Multi-components to Human, Animal and Ecosystem Health: Concepts, Methodologies, and Action Mechanisms is a review well described (although is hard to follow in some sections). This manuscript showed the main characteristics of some probiotics, new methodologies to their characterization and some areas of application. I recommend it for publication after minor changes.
1. The microorganisms’ names should be italicized
2. It would be useful to describe (in a table) some examples of probiotics, prebiotics and common synbiotics
3. Some figures have low resolution
Author Response
Dear Reviewer 2,
Thank you for your pertinent comments/remarks.
Please find below the revisions made to the first version of our manuscript according to your suggestions/comments.
We hope that this revised version will be suitable for the publication of our manuscript to this Special Issue for Microorganisms.
We look forward to hearing from you.
Sincerely,
RAZAFINDRALAMBO Hary (PhD)
Head of ProBioLab
Leader of WP4 - URBANE project (Horizon Europe 2021-2027)
https://urbane-project.eu/Projects/URBANE.html
--------------------------------------------------
TERRA Teaching & Research Centre
Gembloux Agro-Bio Tech
University of Liege
BELGIUM
Email: h.razafindralambo@uliege.be
Phone: +32 81 62 26 23
REVIEWER 2
1. The microorganisms' names are now italicized (throughout the manuscript)
2. A Table listing some synbiotics with pre- and probiotic components, as well as the product/formulation names is added (Table 1): Line 144-145
3. Figs 1-4 have been updated in a better resolution.